# Clinical Characteristics and Predictive Outcomes of Recurrent Nasopharyngeal Carcinoma—A Lingering Pitfall of the Long Latency

**DOI:** 10.3390/cancers14153795

**Published:** 2022-08-04

**Authors:** Yung-Hsuan Chen, Sheng-Dean Luo, Shao-Chun Wu, Ching-Nung Wu, Tai-Jan Chiu, Yu-Ming Wang, Yao-Hsu Yang, Wei-Chih Chen

**Affiliations:** 1Department of Otolaryngology, Kaohsiung Chang Gung Memorial Hospital and Chang Gung University College of Medicine, Kaohsiung 833, Taiwan; 2Department of Anesthesiology, Kaohsiung Chang Gung Memorial Hospital and Chang Gung University College of Medicine, Kaohsiung 833, Taiwan; 3Department of Hematology-Oncology, Kaohsiung Chang Gung Memorial Hospital and Chang Gung University College of Medicine, Kaohsiung 833, Taiwan; 4Department of Radiation Oncology, Kaohsiung Chang Gung Memorial Hospital and Chang Gung University College of Medicine, Kaohsiung 833, Taiwan; 5Department of Traditional Chinese Medicine, Chang Gung Memorial Hospital, Chiayi 613, Taiwan; 6Health Informatics and Epidemiology Laboratory, Chang Gung Memorial Hospital, Chiayi 613, Taiwan; 7School of Traditional Chinese Medicine, College of Medicine, Chang Gung University, Taoyuan 333, Taiwan

**Keywords:** head and neck cancer, relapse, prevalence, salvage surgery, Epstein-Barr virus (EBV) DNA

## Abstract

**Simple Summary:**

Unlike other head and neck cancers, nasopharyngeal carcinoma (NPC) with long-latent recurrence (>five years) is not rare. This study aims to investigate the clinical characteristics and outcomes of long-latent recurrent NPC. We found that 8.1% recurrence occurred after five years. About two thirds of these patients’ primary cancer belonged to the early stages, and almost half of the recurrence was a local recurrence. The five-year overall survival and disease-specific survival rate of long-latent recurrence was 19.7% and 24.5%, respectively. Most long-latent local recurrence was found in unresectable rT3 or rT4 cases, while one-third could not be detected by endoscopy due to recurrence in the skull base. A higher percentage of patients who died after long-latent recurrence were symptomatic with a follow-up interval more than six months, reflecting that delayed detection of recurrence may have a great impact on survival outcome. A long-term follow-up is necessary for early detection of NPC recurrence even after five years of disease-free interval.

**Abstract:**

Purpose: To investigate the clinical characteristics, risk factors, and clinical outcomes of long-latent recurrence (>five years) of nasopharyngeal carcinoma (NPC). Methods: This retrospective study enrolled newly diagnosed NPC patients from the Chang Gung Research Database between January 2007 and December 2019. We analyzed the patients’ characteristics and survival outcomes after recurrence. Results: A total of 2599 NPC patients were enrolled. The overall recurrence rate was 20.5%, while 8.1% of patients had long-latent recurrence (>five years). These patients had a higher percentage of initial AJCC (The American Joint Committee on Cancer) stage I/II (60.5%, *p* = 0.001) and local recurrence (46.5%, *p* < 0.001). Unresectable rT3 and rT4 were found in 60% of patients when recurrence and 30% of local recurrence occurred in the skull base, which could not be detected by the regular endoscopy. The five-year overall survival rate of long-latent recurrence was 19.7%. Alive patients tended to be asymptomatic but have regular follow-ups with the interval less than six months. Multivariate analysis showed age and initial advanced AJCC stages were independent risk factors of death after recurrence. In contrast, patients with recurrence between two and five years, salvage surgeries, and regional recurrence had favorable survival outcomes. Conclusion: Long-latent NPC recurrence is not rare, and the survival outcome is poor. Regular follow-up for early detection of NPC recurrence is necessary even after five years of disease-free period.

## 1. Introduction

Nasopharyngeal carcinoma (NPC) is a common head and neck malignancy with a higher prevalence in endemic areas, including southeast Asia and North Africa [1,2]. Due to a relative insidious site and silent symptoms in the early stage, NPC was usually diagnosed in an advanced stage [3]. In head and neck squamous cell carcinoma (HNSCC), approximately 10% of patients have distant metastasis (DM) at their initial presentation. However, the incidence of DM at the time of diagnosis is higher in nasopharyngeal and hypopharyngeal carcinoma, which could reach 24% in an advanced stage [4]. Fortunately, NPC is a radiosensitive tumor, and radiotherapy or concurrent chemoradiotherapy (CCRT) is the current standard treatment of NPC [1,2]. Compared to other head and neck malignancies, the clinical outcome of NPC is more favorable. The five-year overall survival (OS) rate in the early and the late stages of NPC is around 83–97% and 63–80%, respectively [3,5,6]. On the other hand, the five-year OS of oral SCC ranges from 36% to 57%, and the five-year OS of p16 negative oropharynx SCC ranges from 24% to 48% [7,8]. However, the recurrence rate of NPC remains about 10% to 36% [9,10,11,12] despite that the intensity-modulated radiotherapy (IMRT) with or without chemotherapy evolved as the mainstay treatment of NPC in the past two decades [1,2,3,13]. Moreover, the side effects of chemoradiotherapy are still great concerns to the patients, including mucositis, dysgeusia, xerostomia, dysphagia, hearing loss, and hematological dysfunctions [14].

The management of recurrent NPC could be challenging, and the OS outcome is poor after tumor recurrence [10,15,16]. In the past, re-irradiation was the mainstay of treatment for recurrent NPC. However, the radiation dose of surrounding normal structures is usually close to the tolerance limit in the first treatment course. Therefore, re-irradiation may cause miserable complications and impairments of the quality of life [10,15,16]. Otherwise, relative radioresistant tumors in recurrent NPC may cause treatment failure [10,17,18]. Surgical resection of recurrent NPC is another treatment option [1,2,15,19,20,21]. Compared to re-irradiation, surgical resection has a better outcome in recurrent NPC [19,22]. Liu et al. reported a three-year OS of 85.8% in the endoscopic nasopharyngectomy patients and OS of 68.0% in the IMRT group. Otherwise, less adverse events were found in patients with surgical resection [22]. The anatomic location of the nasopharynx is near the skull base and close to the internal carotid arteries, which causes difficulty in surgical resection with an adequate margin. Due to the advancement of endoscopic instruments and surgical techniques, endoscopic resection of locally recurrent tumors became more popular in recent decades [22,23]. Besides, endoscopic surgery has less severe complications and lower local recurrence rates [24]. Patients who receive the endoscopic nasopharyngectomy for a recurrent tumor less than rT3 stage have a better two-year OS rate, compared to the open surgery [24]. However, surgical resection of recurrent NPC was only indicated for early recurrent tumors or selected T3 patients. [1,2,25]. Therefore, early detection of recurrent tumors is crucial to survival outcomes [26,27,28].

Most tumor recurrences happen within two years in head and neck malignancies and disease relapse after more than three years is rare [9,29,30,31]. The follow-up period from one to six-month intervals within five years is recommended by guidelines [32,33]. However, tumor recurrence after five years is not rare in NPC [9,34,35]. Therefore, a more than five-year follow-up period may be necessary for NPC patients. There were few studies about the clinical characteristics and outcomes of long-latent recurrent NPC. Besides, those studies were old with different treatment strategies from the current guidelines. The purpose of this study was to investigate the clinical characteristics and risk factors of long-latent recurrent NPC. The clinical outcomes of long-latent NPC were also analyzed.

## 2. Materials and Methods

### 2.1. Patient Recruitment

This cohort study was approved by the Institutional Review Board (IRB) of the Kaohsiung Chang Gung Memorial Hospital with reference number 202001131B0. The requirement for informed consent was waived, according to retrospective study design and IRB regulations. From January 2007 to December 2019, 3269 patients diagnosed with nasopharyngeal malignancy in the Chang Gung Research Database were identified by ICD-9 and ICD-10. Exclusion criteria were the initial AJCC Stage IVc at diagnosis, ever diagnosed as NPC before 2007, nasopharyngeal malignancies with morphology codes (SNOMED pathology) other than 8010 (carcinoma), 8020 (undifferentiated carcinoma), 8070 (squamous cell carcinoma), 8071 (keratinizing squamous cell carcinoma), 8072 (non-keratinizing squamous cell carcinoma), and 8082 (lymphoepithelial carcinoma), no curative-intent treatment, and failed to the treatment for primary NPC. According to the recurrent period, the patients with recurrent NPC were divided into three groups, including recurrence within two years, between two and five years, and more than five years (long-latent recurrence). The risk factors of tumor recurrence and the survival outcomes of patients with recurrent tumors were analyzed. The definition of prolonged radiotherapy is the completed initial treatment course more than 56 days. This may be due to any condition, such as adverse effects of radiotherapy and poor compliance, which caused the interruption of primary radiotherapy and led to a longer course of the treatment. The status of recurrence and cause of death in NPC patients was followed by our manager of cancer center and an annual report will be feedback from Health Promotion Administration.

### 2.2. Salvage Treatment

Salvage treatments were decided based on a multidisciplinary team (MDT) discussion for each patient who experienced recurrence. The clinical target volume (CTV) was defined as the recurrent gross tumor and the planning target volume was the CTV with 0.3 cm margin in all directions. The prescribed dose was 54 Gy to 63 Gy in 30 daily fractions delivered by intensity-modulated radiotherapy during this study period. Salvage surgeries included the radical neck dissection for regional recurrence and the nasopharyngectomy for local recurrence. The chemotherapy regimens for recurrent NPC was mainly cisplatin plus 5-fluorouracil (5-FU), also known as the PF regimen, or carboplatin.

### 2.3. Statistical Analysis

The correlations between the continuous variables and recurrence or death status were tested by the independent *t*-test, while the correlations between the categorical variables and recurrence or death status were tested by either a two-sided Fisher’s exact test or a Pearson’s chi-squared test. Variables included age, sex, comorbidities, T classification, N classification, AJCC stages of cancer, Epstein-Barr virus (EBV) DNA status, treatment protocol, duration of radiotherapy, WHO pathological types, and types of recurrence. The differences in clinicopathological variables among recurrence groups of different latency were analyzed by the one-way ANOVA test. Logistic regression and Cox regression were used to evaluate the clinical outcomes, and multivariate analysis was also performed to adjust possible confounders (e.g., age, sex, comorbidities, clinical tumor characteristics, and treatment). The Kaplan-Meier method was used for the survival analysis, and the log-rank test was used to check the survival function. All statistical analyses were performed using SAS version 9.4 (SAS Institute, Cary, NC, USA). Statistical significance was set for each analysis at a *p*-value < 0.05. The five-year overall survival (OS) is defined as the percentage of patients who have not died five years after the diagnosis of the recurrent NPC. The patients who died from other causes will be counted as events. The disease-specific survival (DSS) is defined as the percentage of patients who have not died from NPC. The patients who died from other causes will not be counted as events.

## 3. Results

### 3.1. Study Population

The cohort consisted of 3269 patients diagnosed with nasopharyngeal malignancy in Chang Gung Medical Foundation, Taiwan, between January 2007 and December 2019. Among the 3269 patients, 446 patients were not recruited after applying the exclusion criteria, including AJCC Stage IVc at the initial diagnosis (*n* = 267), ever diagnosed as NPC before 2007 (*n* = 9), nasopharyngeal malignancies with the SNOMED morphology codes other than 8010, 8020, 8070, 8071, 8072, and 8082 (*n* = 81), and no curative-intent treatment (*n* = 89). Besides, we excluded 224 patients who never achieved complete remission for at least 6-month intervals after treatment. Finally, 2599 NPC patients post successful treatment were enrolled in our study. Figure 1 shows a flow chart of the cohort study design for statistical analysis.

### 3.2. Clinical Characteristics and Time Distribution of Recurrent NPC

The overall recurrence rate was 20.5% (533/2599). Demographics and clinical characteristics of patients with recurrent and non-recurrent NPC are summarized in Table 1. Compared to the non-recurrence group, the patients with recurrent NPC are more likely to be males (79.9% vs. 73.9%, *p* = 0.004), have advanced initial clinical T classification (56.5% vs. 42.7%, *p* < 0.001), N classification (91.2% vs. 82.3%, *p* < 0.001), and AJCC stage at diagnosis (83.9% vs. 64.3%, *p* < 0.001), longer radiotherapy period (31.0% vs. 21.2%, *p* < 0.001), higher death rate (66.4% vs. 11.8%, *p* < 0.001), receive CCRT as the initial treatment (84.6% vs. 76.1%, *p* < 0.001), and undifferentiated carcinoma (33.0% vs. 24.4%, *p* < 0.001). The logistic regression showed that the patients with recurrent NPC are more likely to be males [odds ratio [OR], 1.35; 95% confidence interval [CI], 1.07–1.72; *p* = 0.013], have advanced initial AJCC stages at diagnosis (OR, 2.64; 95% CI, 2.04–3.42; *p* < 0.001), and longer radiotherapy period (OR, 1.52; 95% CI, 1.22–1.89; *p* < 0.001) in adjusted models (Table 2). The time distribution of recurrent NPC is shown in Figure 2, wherein 315 (59.1%) patients had recurrence within two years, 175 (32.8%) had tumor recurrence from two to five years, and 43 (8.1%) patients had long-latent tumor recurrence (more than five years). Demographics and clinical characteristics among patients with different tumor recurrence latency are summarized in Table 3. The patients with long-latent tumor recurrence had a higher percentage of early AJCC stages (≤2 years vs. 2–5 years vs. >5 years, 11.7% vs. 20.6% vs. 30.2%, *p* = 0.001), T classification (37.5% vs. 50.3% vs. 60.5%, *p* = 0.002), N classification (7.3% vs. 8.6% vs. 20.9%, *p* < 0.001), and local recurrence (30.5% vs. 33.1% vs. 46.5%, *p* < 0.001).

### 3.3. Survival Outcome of Patients with Recurrent NPC

The five-year OS and DSS rate of patients with recurrent NPC were 25.0% and 32.5%, respectively. The patients with recurrence between two and five years had better OS, DSS, and longer median survival time compared to other groups (Figure 3). The Cox regression showed age and advanced AJCC stages were independent risk factors for death after tumor recurrence in both crude and adjusted models. In contrast, patients with regional failure, recurrence between two and five years, and salvage surgeries had better survival outcomes after tumor recurrence (Table 4).

### 3.4. The Long-Latent Recurrence NPC Group

There were 43 patients with long latent tumor recurrence and the median recurrent period was 6.1 years (ranging from 5.0 to 10.1 years). The five-year OS rate and DSS rate of these patients were 19.7% and 24.5%, respectively. A total of 46.5% (20/43) patients had recurrence as a local failure. The patients who died of the disease had a higher percentage of a prolonged follow-up period (36% vs. 16.7%, *p* = 0.163) compared to alive patients but without statistical significance. Twenty-two patients presented with clinical symptoms when recurrence, including neck masses (5), ocular symptoms (4), neurologic symptoms (3), pulmonary symptoms (7), headache (1), bone fracture (1), and nasal congestion (1). Patients who died of disease were more likely to have clinical symptoms when recurrence (68.0% vs. 27.8%, *p* = 0.014). Other 21 patients were asymptomatic but diagnosed with recurrent NPC by regular examinations. A higher percentage of alive patients was asymptomatic (72.2% vs. 32.0%, *p* = 0.014) and received regular follow-up within six months (83.3% vs. 64.0%, *p* = 0.163) (Table 5). In patients with local recurrence, 30% (6/20) patients had normal mucosa appearance under scope examination. The local recurrence occurred in the skull base region, which could not be detected by the endoscopy.

## 4. Discussion

Attributing to the early detection with EBV-related biomarkers [26,36], the application of IMRT [37], and the establishment of clinical guidelines [38], the treatment outcome of nasopharyngeal carcinoma has improvements in the past two decades [3]. However, recurrent NPC remains common, with the recurrence rate around 12% to 22% in previous reports [9]. As with other head and neck cancers, most NPC recurrence occur within two years. However, unlike other head and neck cancers, long-latent recurrence (>five years) is not rare in NPC. Lee et al. reported 9% of recurrent NPC patients with long latency [35], while Li et al. reported the percentage of long-latent recurrence was as high as 17.1% [9]. In our study, the overall recurrence rate was 20.5%, and 8.1% of recurrent NPC patients had tumor recurrence five years after treatment. The recurrence rate was compatible with previous reports, but the percentage of long-latent recurrence was lower.

Our study showed the patients with long-latent recurrence had a higher percentage of early AJCC stages and local recurrence. In contrast, more than half of our patients with recurrence within two years suffered from distant metastasis. These findings may be owing to a relative indolent tumor behavior in NPC with long-latent recurrence. Previous studies reported similar results, that long-latent relapse occurred in patients with early T-stages at the initial diagnosis [9,35]. However, Lee et al. reported a different observation that patients with long-latent recurrence had a lower percentage of local recurrence but a higher percentage of distant metastasis [35]. The main difference between Lee’s study and our study cohort lies in the treatment strategy. Their patients received megavoltage radiotherapy without additional chemotherapy from 1976 to 1985. In contrast, most of our patients received IMRT with concurrent chemotherapy according to National Comprehensive Cancer Network (NCCN) guidelines [38]. Nowadays, concurrent chemotherapy is proven to be superior to conformal radiotherapy (CRT) regarding increased overall survival [1], locoregional control rate, disease-free survival, and metastasis-free survival [39].

Various hypotheses explain the occurrence of late recurrence, including indolent tumor behavior, new de novo tumor, and radiation-induced malignancy. First, nasopharyngeal carcinoma is a relatively slow-growing tumor compared to other HNSCC. The median tumor doubling time of HNSCC was around 94–99 days [40,41,42]. In contrast, the median tumor doubling time of NPC was around 279 days, which was three-fold of head and neck SCC [43]. Therefore, we could expect a longer recurrent latency in NPC. Besides, Lee et al. and Nicholls et al. suggested that some relapsed NPC were new growth rather than recurrence [35,44], while some pathological and genetic differences between primary and recurrent NPC were reported [45]. Another possibility is radiation-induced secondary malignancy. The IMRT and the combination with chemotherapy had been reported to increase the incidence of radiation-induced malignancy for NPC patients [46,47]. Radiation-induced squamous cell carcinoma, sarcoma, or osteosarcoma in the nasopharynx or skull base area had been reported, while the median latency from initial radiotherapy to the development of second primary tumor was around 10.5–12.4 years [47,48,49]. Chan et al. suggested the risk factors of developing post-radiation malignancies in NPC patients included the high dose of radiation, young age, and early primary stage with a long subsequent survival [47].

Although nasopharyngeal carcinoma has a favorable outcome of five-year overall survival around 70–90% after primary treatment [6,12,39], the prognosis after tumor recurrence is poor. The five-year overall survival rate of recurrent NPC variates from 11% to 56%, depending on the recurrent staging and the available salvage treatments [15,50,51,52]. In our study, the five-year OS and DSS rate of patients with recurrent NPC were 25.0% and 32.5%, respectively. Since 83.9% of our patients suffered from an advanced stage of recurrent disease, this might lead to a worse survival rate than in previous studies. Newton et al. suggested that the recurrent T stage was associated with the OS, with the five-year OS rate being 60% for rT1-2 and 30% for rT3-4, respectively [15]. Other prognostic factors related to decreased survival rate were elder age, male gender, advanced N stage, and a high baseline value of plasma EBV DNA [34,53,54]. Our study showed a similar result that age and initial advanced AJCC stage were independent risk factors for death after tumor recurrence. By contrast, tumor recurrence between two and five years, regional recurrence, and salvage surgery were the independent prognostic factors for better survival outcomes. In the past, salvage radiotherapy was the mainstay of treatment for patient with recurrent NPC. However, more radioresistant recurrent tumors cause worse treatment results [10,17,18]. Therefore, salvage surgeries for resectable locoregional recurrence have been advocated in recent years [1,2,15,21]. Zhang et al. reported that patients receiving salvage neck dissection for regional recurrence had a better survival rate than those receiving repeated radiotherapy [19]. Nasopharyngectomy for resectable locally recurrent NPC also showed a favorable outcome, with the overall survival rate around 85% in patients receiving endoscopic nasopharyngectomy and 68% in re-irradiation patients [22,55]. We also found that patients with recurrence between two and five years had better OS, DSS and a longer median survival time. The possible explanation for this finding is that patients usually received regular follow-up and examinations within five years so that the recurrence could be detected earlier and treated. The HNSCCs were regarded as cured after five years of disease-free interval. Therefore, the patients and clinicians may have less awareness of disease recurrence or even cease regular follow-up as the five-year follow-up period with variable intervals is widely used [29]. However, based on the finding of our study, we advocate a continuing regular follow-up program with less than six months interval for patients with NPC even after a five-year disease-free interval.

In our cohort, the five-year overall and disease-specific survival rates of long-latent recurrent patients were 19.7% and 24.5%, respectively. Though nearly half of patients with long-latent recurrence belonged to local recurrence, almost two-thirds were unresectable rT3 or rT4 cases. Besides, one-third of long-latent locally recurrent patients had normal mucosa appearance under endoscopy due to relapse in the skull base region. Lee et al. reported that a high proportion of long-latent recurrent NPC had extended outside the nasopharynx [35]. Li et al. found that 46.9% of recurrent tumors belonged to more advanced stages, and half occurred in the margin or outside of the primary tumor [56]. The areas outside the primary tumor received lower radiation doses. Therefore, the subclinical tumor cells within such low-dose areas could survive and lead to tumor recurrence [9]. We also found that a significantly substantial proportion of patients who died after recurrence were symptomatic and tended to have prolonged follow-up periods. Therefore, regular follow-up at least five years after radiotherapy is vital. The endoscopic examination alone may not be sufficient, and the periodic MRI is essential for detecting subclinical or outfield recurrence [57,58,59]. However, MRI is expensive, while the phantom tumor phenomenon is not uncommon in MRI, particularly after completion of radiotherapy more than five years [60]. Recently the plasma EBV DNA has been regarded as a valuable tool for early detection of either primary surveillance or recurrence [61,62,63]. Xu et al. found that the increased EBV DNA titer was present at a median of 2.6 months before radiographic progression [64]. Furthermore, the cost of the EBV DNA titer is much less expensive than the MRI of the nasopharynx. Therefore, regular follow-up of the plasma EBV DNA may be a cost-effective method for long-term follow-up.

There were a few limitations in this study. As a retrospective cohort study, there might be some potential selection bias. The plasma EBV DNA level was not analyzed in this study due to most patients diagnosed before 2012 had no EBV DNA data. The EBV DNA test was not widely available and the test fee was not covered by the National Health Insurance (NHI) before 2012. The CGRD had a significantly higher severity of comorbidities and prevalence of specific diseases than those in the Taiwan National Health Insurance Research Database (NHIRD), which could lead to over-estimate the disease-severity of NPC in Taiwan [65]. Additionally, though the recurrence rate in our study is similar to previous studies, some patients with NPC recurrence may be diagnosed in other hospitals. It might cause an underestimation of the recurrence rate or death. However, we filed the patients’ medical records in our cancer registry database after the diagnosis of the primary NPC, and our oncology case managers are in charge of tracing and updating the patients’ conditions. Besides, the Health Promotion Administration in Taiwan will feedback an annual report of the departed patients and the causes of death. Therefore, we still have information about the recurrence or death even if the patient did not receive further medical services in any of the Chang Gung Hospitals. Last but not least, since Taiwan is one of the endemic areas for NPC, the result of our study may not be suitable to be applied in other countries with a low prevalence of NPC.

## 5. Conclusions

The present study shows that long-latent recurrence occurred in 8.1% of recurrent NPC. The patients with long-latent tumor recurrence had a higher percentage of initial AJCC stage I/II and local recurrence. However, most long-latent local recurrences were unresectable rT3 or rT4 cases, leading to a poor survival outcome. The patients who died after long-latent recurrence were usually symptomatic and had a prolonged follow-up period (more than six months). Even under regular endoscopic examinations, we found that 30% of patients had a long-latent recurrence in the skull base, which could not be detected. Early detection and treatment might have a substantial impact on survival outcomes. Therefore, regular follow-up with various examinations, even after a five-year disease-free interval, is imperative for the early detection of recurrent NPC.

## Figures and Tables

**Figure 1 cancers-14-03795-f001:**
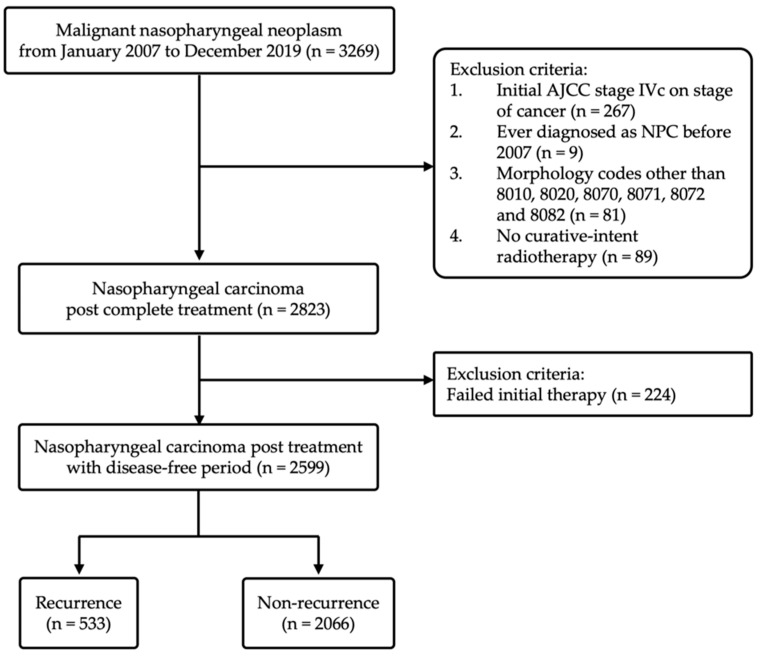
Flow chart of the cohort study design.

**Figure 2 cancers-14-03795-f002:**
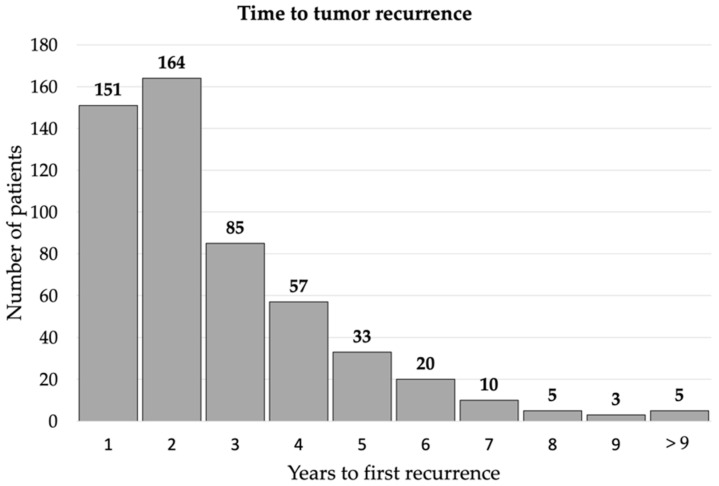
Time distribution of tumor recurrence: 315 (59.1%) patients had recurrence within 2 years, 175 (32.8%) had tumor recurrence from 2 to 5 years, and 43 (8.1%) patients had long-latent tumor recurrence (more than five years).

**Figure 3 cancers-14-03795-f003:**
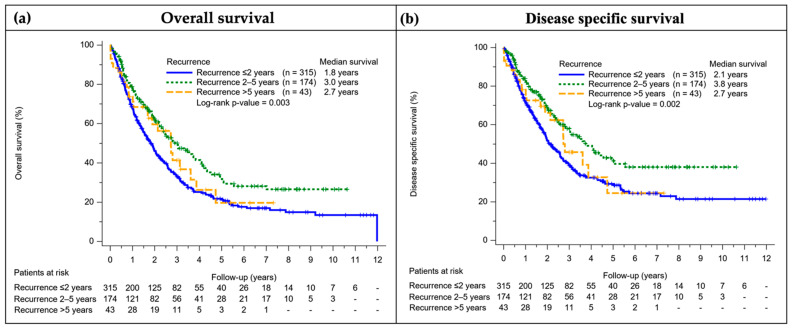
Survival curve of recurrent groups with different time intervals: the patients with recurrence between two and five years had better overall survival (**a**), disease specific survival (**b**), and longer median survival time compared to other groups. The Kaplan-Meier method was used for the survival analysis, and the log-rank test was used to check the survival function.

**Table 1 cancers-14-03795-t001:** Demographic data (recurrence vs. non-recurrence).

Variables	Recurrence (*n* = 533)	Non-Recurrence (*n* = 2066)	*p*-Value
**Age** (Mean ± SD, Years) (*n* = 2599)	50.2 ± 12.1	49.9 ± 12.2	0.512
**Sex** (*n* = 2599)			0.004
Male	426 (79.9%)	1526 (73.9%)	
Female	107 (20.1%)	540 (26.1%)	
**Clinical T classification** (*n* = 2599)			<0.001
1, 2	232 (43.5%)	1183 (57.3%)	
3, 4	301 (56.5%)	883 (42.7%)	
**Clinical N classification** (*n* = 2599)			<0.001
0	47 (8.8%)	365 (17.7%)	
1, 2, 3	486 (91.2%)	1701 (82.3%)	
**Clinical AJCC stage** (*n* = 2599)			<0.001
1, 2	86 (16.1%)	738 (35.7%)	
3, 4	447 (83.9%)	1328 (64.3%)	
**Death** (*n* = 2599)	354 (66.4%)	243 (11.8%)	<0.001
**Diabetes mellitus** (*n* = 2599)	20 (3.8%)	108 (5.2%)	0.161
**Hypertension** (*n* = 2599)	40 (7.5%)	190 (9.2%)	0.220
**Prolonged RT duration** (*n* = 2599)			<0.001
No	368 (69.0%)	1629 (78.8%)	
Yes	165 (31.0%)	437 (21.2%)	
**Treatment protocol** (*n* = 2599)			<0.001
RT alone	35 (6.6%)	287 (13.9%)	
CCRT	451 (84.6%)	1572 (76.1%)	
Induction CT + CCRT	47 (8.8%)	207 (10.0%)	
**WHO type** (*n* = 2599)			<0.001
Keratinizing squamous cell carcinoma	57 (10.7%)	174 (8.4%)	
Non-keratinizing carcinoma, differentiated	300 (56.3%)	1387 (67.1%)	
Non-keratinizing carcinoma, undifferentiated	176 (33.0%)	505 (24.4%)	
**Recurrence type** (*n* = 2599)			
Local	174 (32.6%)	-	
Regional	122 (22.9%)	-	
Distal	237 (44.5%)	-	

AJCC = American Joint Committee on Cancer; RT = radiotherapy; CT = Chemotherapy; CCRT = Concurrent chemoradiotherapy; WHO = World Health Organization.

**Table 2 cancers-14-03795-t002:** Logistic regression of independent risk of recurrent NPC.

	Overall (*n* = 2599)			
Variables	Crude OR (95% C.I.)	*p*-Value	Adjusted OR (95% C.I.)	*p*-Value
**Age** (Mean ± SD) (years) (*n* = 2599)	1.00 (0.99, 1.01)	0.512	1.01 (1.00, 1.01)	0.148
**Sex** (*n* = 2599)				
Male	1.41 (1.12, 1.78)	0.004	1.35 (1.07, 1.72)	0.013
Female	1.00		1.00	
**AJCC** (*n* = 2599)				
1, 2	1.00		1.00	
3, 4	2.89 (2.25, 3.70)	<0.001	2.64 (2.04, 3.42)	<0.001
**Diabetes mellitus** (*n* = 2599)				
No	1.00		1.00	
Yes	0.71 (0.43, 1.15)	0.163	0.78 (0.45, 1.34)	0.362
**Hypertension** (*n* = 2599)				
No	1.00		1.00	
Yes	0.80 (0.56, 1.14)	0.221	0.90 (0.60, 1.35)	0.602
**Prolonged RT duration** (*n* = 2599)				
No	1.00		1.00	
Yes	1.67 (1.35, 2.07)	<0.001	1.52 (1.22, 1.89)	<0.001
**Treatment protocol** (*n* = 2599)				
RT alone	0.54 (0.33, 0.86)	0.010	0.78 (0.48, 1.29)	0.336
CCRT	1.26 (0.91, 1.76)	0.169	1.27 (0.90, 1.78)	0.177
Induction CT + CCRT	1.00		1.00	
**WHO type** (*n* = 2599)				
Keratinizing squamous cell carcinoma	1.00		1.00	
Non-keratinizing carcinoma, differentiated	0.66 (0.48, 0.91)	0.012	0.70 (0.50, 0.98)	0.038
Non-keratinizing carcinoma, undifferentiated	1.06 (0.75, 1.50)	0.724	1.13 (0.79, 1.61)	0.508

AJCC = American Joint Committee on Cancer; RT = radiotherapy; CCRT = Concurrent chemoradiotherapy; CT = Chemotherapy; WHO = World Health Organization.

**Table 3 cancers-14-03795-t003:** Demographic data of recurrent groups with different time intervals.

Variables	Recurrence ≤ 2 Years (*n* = 315)	Recurrence 2–5 Years (*n* = 175)	Recurrence > 5 Years (*n* = 43)	*p*-Value
**Age** (Mean ± SD) (years) (*n* = 533)	50.4 ± 12.1	50.1 ± 12.3	49.5 ± 12.1	0.896
**Sex** (*n* = 533)				0.962
Male	253 (80.3%)	139 (79.4%)	34 (79.1%)	
Female	62 (19.7%)	36 (20.6%)	9 (20.9%)	
**T** (*n* = 533)				0.002
1, 2	118 (37.5%)	88 (50.3%)	26 (60.5%)	
3, 4	197 (62.5%)	87 (49.7%)	17 (39.5%)	
**N** (*n* = 533)				0.013
0	23 (7.3%)	15 (8.6%)	9 (20.9%)	
1, 2, 3	292 (92.7%)	160 (91.4%)	34 (79.1%)	
**AJCC** (*n* = 533)				0.001
1, 2	37 (11.7%)	36 (20.6%)	13 (30.2%)	
3, 4	278 (88.3%)	139 (79.4%)	30 (69.8%)	
**Death** (*n* = 533)	232 (73.7%)	97 (55.4%)	25 (58.1%)	<0.001
**Diabetes mellitus** (*n* = 533)	12 (3.8%)	5 (2.9%)	3 (7.0%)	0.443
**Hypertension** (*n* = 533)	24 (7.6%)	12 (6.9%)	4 (9.3%)	0.856
**Prolonged RT duration** (*n* = 533)				0.972
No	218 (69.2%)	121 (69.1%)	29 (67.4%)	
Yes	97 (30.8%)	54 (30.9%)	14 (32.6%)	
**Treatment protocol** (*n* = 533)				0.432
RT alone	17 (5.4%)	13 (7.4%)	5 (11.6%)	
CCRT	271 (86.0%)	147 (84.0%)	33 (76.7%)	
Induction CT + CCRT	27 (8.6%)	15 (8.6%)	5 (11.6%)	
**WHO type** (*n* = 533)				0.980
Keratinizing squamous cell carcinoma	34 (10.8%)	19 (10.9%)	4 (9.3%)	
Non-keratinizing carcinoma, differentiated	179 (56.8%)	98 (56.0%)	23 (53.5%)	
Non-keratinizing carcinoma, undifferentiated	102 (32.4%)	58 (33.1%)	16 (37.2%)	
**Recurrence type** (*n* = 533)				<0.001
Local	96 (30.5%)	58 (33.1%)	20 (46.5%)	
Regional	56 (17.8%)	55 (31.4%)	11 (25.6%)	
Distal	163 (51.7%)	62 (35.4%)	12 (27.9%)	
**Salvage surgery** (*n* = 533)				0.527
No	270 (85.7%)	144 (82.3%)	35 (81.4%)	
Yes	45 (14.3%)	31 (17.7%)	8 (18.6%)	

AJCC = American Joint Committee on Cancer; RT = radiotherapy; CT = Chemotherapy; CCRT = Concurrent chemoradiotherapy; WHO = World Health Organization.

**Table 4 cancers-14-03795-t004:** Cox regression analysis of survival after tumor recurrence.

Variables	Univariable HR (95% C.I.)	*p*-Value	Multivariable HR (95% C.I.)	*p*-Value
**Age** (years)	1.03 (1.02, 1.04)	<0.001	1.03 (1.02, 1.04)	<0.001
**Sex**				
Male	0.97 (0.75, 1.27)	0.847	1.01 (0.77, 1.33)	0.929
Female	1.00		1.00	
**AJCC**				
1, 2	1.00		1.00	
3, 4	1.56 (1.14, 2.12)	0.005	1.46 (1.04, 2.04)	0.027
**Diabetes mellitus**				
No	1.00		1.00	
Yes	1.82 (1.10, 3.00)	0.020	1.41 (0.81, 2.46)	0.225
**Hypertension**				
No	1.00		1.00	
Yes	1.34 (0.92, 1.96)	0.129	0.92 (0.60, 1.40)	0.690
**Prolonged RT duration**				
No	1.00		1.00	
Yes	1.30 (1.05, 1.62)	0.017	1.16 (0.93, 1.46)	0.190
**Treatment protocol**				
CCRT	1.00		1.00	
RT alone	1.38 (0.92, 2.05)	0.118	1.38 (0.89, 2.13)	0.149
**WHO type**				
Keratinizing squamous cell carcinoma	1.00		1.00	
Non-keratinizing carcinoma, differentiated	0.91 (0.64, 1.30)	0.617	0.81 (0.57, 1.16)	0.247
Non-keratinizing carcinoma, undifferentiated	0.97 (0.68, 1.39)	0.868	0.93 (0.65, 1.35)	0.719
**Recurrence**				
Recurrence ≤ 2 years	1.00		1.00	
Recurrence 2–5 years	0.66 (0.52, 0.84)	0.001	0.69 (0.54, 0.88)	0.003
Recurrence > 5 years	0.82 (0.54, 1.24)	0.344	0.91 (0.59, 1.39)	0.665
**Recurrence type**				
Local	1.00		1.00	
Regional	0.70 (0.51, 0.95)	0.023	0.72 (0.52, 0.99)	0.046
Distal	1.45 (1.14, 1.83)	0.002	1.05 (0.81, 1.34)	0.725
**Salvage surgery**				
No	1.00		1.00	
Yes	0.30 (0.21, 0.44)	<0.001	0.38 (0.25, 0.57)	<0.001

HR = Hazard ratios; AJCC = American Joint Committee on Cancer; RT = radiotherapy; CT = chemotherapy; CCRT = Concurrent chemoradiotherapy; WHO = World Health Organization.

**Table 5 cancers-14-03795-t005:** Demographic data of long-latent recurrent group (alive vs. dead).

Variables	Dead (*n* = 25)	Alive (*n* = 18)	*p*-Value
**Age** (Mean ± SD) (years) when recurrence (*n* = 43)	58.5 ± 11.3	54.6 ± 13.3	0.304
**Sex** (*n* = 43)			0.712
Male	19 (76.0%)	15 (83.3%)	
Female	6 (24.0%)	3 (16.7%)	
**T classification when recurrence** (*n* = 43)			0.003
0	5 (20.0%)	11 (61.1%)	
1, 2	7 (28.0%)	6 (33.3%)	
3, 4	13 (52.0%)	1 (5.69%)	
**N classification when recurrence** (*n* = 43)			0.158
0	19 (76.0%)	10 (55.6%)	
1, 2, 3	6 (24.0%)	8 (44.4%)	
**AJCC stage when recurrence** (*n* = 43)			0.006
1, 2	4 (16.0%)	10 (55.6%)	
3, 4	21 (84.0%)	8 (44.4%)	
**Prolonged RT duration** (*n* = 43)			0.059
No	14 (56.0%)	15 (83.3%)	
Yes	11 (44.0%)	3 (16.7%)	
**WHO type when recurrence** (*n* = 43)			0.311
Keratinizing squamous cell carcinoma	1 (4%)	2 (33.3%)	
Non-keratinizing carcinoma, differentiated	7 (28.0%)	4 (22.2%)	
Non-keratinizing carcinoma, undifferentiated	9 (36.0%)	10 (66.7%)	
Others	8 (32.0%)	2 (11.1%)	
**Recurrent type** (*n* = 43)			0.036
Local	15 (60.0%)	5 (27.8%)	
Regional	3 (12.0%)	8 (44.4%)	
Distal	7 (28.0%)	5 (27.8%)	
**Salvage surgery** (*n* = 43)			0.247
No	22 (88.0%)	13 (72.2%)	
Yes	3 (12.0%)	5 (27.8%)	
**Symptomatic** (*n* = 43)			0.014
Yes	17 (68.0%)	5 (27.8%)	
No	8 (32.0%)	13 (72.2%)	
**Prolonged follow-up interval** **(>6 months)** (*n* = 43)			0.163
Yes	9 (36.0%)	3 (16.7%)	
Nos	16 (64.0%)	15 (83.3%)	

AJCC = American Joint Committee on Cancer; RT = radiotherapy; WHO = World Health Organization.

## Data Availability

The data presented in this study are available in this article.

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
