# Peer review of "Clinical Characteristics and Predictive Outcomes of Recurrent Nasopharyngeal Carcinoma—A Lingering Pitfall of the Long Latency"

_cancers, 2022, doi:10.3390/cancers14153795_

Round 1
Reviewer 1 Report
- In the title I should add clinical charateristics and predictive outcomes of....
- line 48, frequently head and neck cancer is already metastatic at diagnosis, with a poorer prognosis. Please discuss and cite doi:10.14639/0392-100X-suppl.1-40-2020.
- line 55, head and neck tumours, especially in relation to HPV or EBV infection could manifest a different prognosis, leading to a different primary treatment. Chemioradiotherapy is considered the first line treatment but common overall side effects are possible as dysphagia or hearing loss. please discuss and cite doi:10.1016/j.anl.2021.05.007.
- line 67, Consensus has yet to be reached on the optimal operation for patients with residual or recurrent nasopharyngeal carcinoma (NPC). Endoscopic surgeries achieved bigger rates than open surgeries in patients with recurrent tumor (rT) 1, rT2, and rT3 (93% vs 87%, 77% vs 63%, 67% vs 53%). As for patients with rT4, 2-year OS was similar (35% vs 35%).In addition, the former is less severe complications, lower local recurrence rates (27% vs 32%). please discuss and cite doi:10.1002/hed.26397.
- strobe guidelines should be applied in the methods to improve the structure
- the consort model diagram could be adapted the present.
Discussion
- endoscopic approach offers a safe and efficient alternative to open approach with better short-term outcome and fewer postoperative complications in selecting patients strictly. A systematic search of Pubmed/Medline, Embase, and Cochrane databases ranged between 2000 and 2017 was conducted, including studies reporting specific residual or local recurrent nasopharyngeal cancer survival data., please discuss and cite doi:10.3760/cma.j.issn.1673-0860.2019.09.006.
Reviewer 2 Report
1- English: The manuscript could benefit from editing for grammar, missing words, and subject-verb agreement, etc. It is recommended that authors delete irrelevant "general" phrases and sentences, repeated and unneeded words. They should use short sentences. Also, some Introductory sentences are irrelevant or are not needed. There are also typos in the manuscript.
2- Abbreviations: All abbreviations should be revised and defined at their first use.
3- References: Many old references used need to be updated.
4- Simple summary: This statement needs rephrasing “We found that 8.1% recurrence occurred after five…”
5- Simple summary: it is not clear what is meant by long-latent recurrence. Do authors mean recurrence that occurs after more than 5 years? If so, please define this term clearly as all the manuscript revolves around it. Also, what is long-latent local recurrence?
6- Abstract: “These patients had a higher percentage of initial AJCC stage I/II (60.5%) and local recurrence (46.5%).” Was this statistically significant?
7- Abstract: “Unresectable rT3 and rT4 were found in 60% of patients.” This staging is the initial staging at diagnosis or at recurrence?
8- Abstract: “Alive patients tended to be asymptomatic but had regular follow-ups within six months.” Follow up was only within 6 months of diagnosis or every 6 months since diagnosis?
9- Abstract: “Regular follow-up for early detection of NPC recurrence is necessary even after five years of disease-free period.” This conclusion is based on what results the authors found? Do patients stop follow up after 5 years usually?
10- Introduction: “NPC is a radiosensitive tumor [4], and radiotherapy or chemoradiotherapy is the current standard treatment of NPC.” Treatment is radiotherapy and/or chemotherapy. Please correct.
11- Introduction: “Compared to other head and neck malignancies, the clinical outcome of NPC is more favorable.” Give examples.
12- Introduction: “despite the intensity-modulated radiotherapy with or without chemotherapy evolved as the mainstay treatment of NPC in the past two decades.” This statement needs rephrasing: “despite that the intensity-modulated radiotherapy with or without chemotherapy evolved as the mainstay treatment of NPC in the past two decades.”
13- Introduction: “Compared to re-irradiation, surgical resection has a better overall survival rate in recurrent NPC, though selection bias may exist.” Please give examples and numbers showing how surgery after recurrence improves overall survival.
14- Methods: what are the inclusion criteria?
15- Methods: Exclusion criteria need to be clarified: “Exclusion criteria were AJCC Stage IVc, ever diagnosed as NPC before 2007, nasopharyngeal malignancies with morphology codes (SNOMED pathology) other than 8010, 8020, 8070, 8071, 8072, and 8082, no curative-intent treatment, and failed to the treatment for primary NPC.” The numbers 8010, 8020, etc. are the ICD codes which need to be clarified as so. Also, what about having other malignancies which might affect results? Were those patients also excluded? AJCC Stage IVc is the initial staging at diagnosis?
16- Methods: which edition of AJCC was used? We know that many of those AJCC guidelines are updated every couple of years. If patients included are between 2007 and 2019, were patients staged according to old guidelines and then restaged for the purpose of this study? This needs to be clarified and mentioned clearly.
17- Methods: what about the M stage? Are all patients included metastasis-free at the time of diagnosis. Also, if EBV status is collected, why isn't it mentioned anywhere in the results, Tables, or Discussion?
18- Results: “morphology codes other than 8010, 8020, 8070, 8071, 8072, 8082.” Are those the ICD codes or morphology codes?
19- Results: in Figure 1, dates formatting 2007.01 and 2019.12 should be made clearer: January 2007 to December 2019. Also, “Failed to curative treatment” should be rephrased: “Failed initial therapy.”
20- Results: “The logistic regression showed the patients with recurrent NPC had a higher percentage of male, advanced AJCC stages, prolonged radiotherapy period, and histological type as undifferentiated carcinoma.” The terms used in describing the results are not scientifically sound. This statement for example should have been written as: “The logistic regression showed that the patients with recurrent NPC are more likely to be males, have advanced initial AJCC stage at diagnosis, longer radiotherapy period, and have undifferentiated carcinoma as the histological type.” Also, authors should mention that this difference between recurrence and non-recurrence group of patients was statistically significant and mention the p-values of each. In relation to this, please add the ORs to the results section for a better presentation of results.
21- Results: In Table 1, please add next to Age (mean +/- SD).
22- Results: Table 1, is the T and N stage the pathologic or clinical staging? Please clarify.
23- Results: Table 1, I see that there was statistically significant difference between both groups regarding T and N stages, death, and treatment protocol; however, authors did not mention this in the results section. Is there a reason for not mentioning that?
24- Results: How about the initial tumor size and margin status? Did authors include this in their analysis? Also, how many patients underwent surgery after initial diagnosis? And what specific type of therapy did the patients receive (what chemotherapy and what dose of radiation)? All this need to be included since it could affect the analysis. If a patient received aggressive radiation therapy versus a patient who did not, we expect to see recurrence in the patient with less aggressive treatment. As such, treatment protocol should be standardized among patients and results then should be adjusted for the treatment protocol to avoid bias.
25- Results: in table 2, the word reference should be replaced by the number 1, which is how logistic regression results are usually presented.
26- Results: Figure 2: change “Case number” to “number of patients.”
27- Results: Figure 2: change “years” to “years to first recurrence.”
28- Results: Table 3: “In Brief, the patients with long-latent tumor recurrence had a higher percentage of early AJCC stages and local recurrence.” Be more specific and narrate the results with the p-values in the results section. Also, add mean +/- SD to Age as in Table 1.
29- Results: please define “5-year overall survival (OS)” and “disease-specific survival (DSS).” What about loss of follow-up? This should be included.
30- Results: “A higher percentage of patients presented with clinical symptoms when recurrence was died of disease.” English language used here needs revision. The whole statement needs rephrasing.
31- Results: In the survival analysis, did authors adjust their results for the different parameters included to avoid bias?
32- Figures: All the figure legends can be revised as to be more informative of the images presented. Also, statistical tests used and meaning of asterix need to be added. Abbreviations used withing Tables and Figures should be defined as well in the legends at the end.
33- Tables: Where there missing data for any of the parameters? Total number of patients for each parameter need to be added next to the variable name to make this clear.
34- Tables: Formatting of Tables needs to be revised. Titles of variables can be in bold, and their categories below them not bolded. Authors can refer to this paper to have an idea how to better represent their results: Tables 1, 2, and 3 in https://linkinghub.elsevier.com/retrieve/pii/S1092-9134(21)00024-1. Same applies to Table 2.
35- General comment on results: Authors did not mention whether their results are adjusted for confounding factors. If not, this is a major limitation in the study and makes all results not reliable.
36- Discussion: Authors should focus more on the main findings and avoid repeating results presentation in the discussion. Authors could also correlate their findings with what has been published in literature. Clinical relevance should be added.
Reviewer 3 Report
This is an interesting study about clinical characteristics and survival outcome of long latent recurrent nasopharyngeal carcinoma (NPC). A total of 2599 NPC patients were enrolled.
The paper is well written. However, some issues remain.
Please add EBV-DNA status in the analyses.
The authors should describe the type of salvage surgery and/or the features of re-irradiation. What about chemotherapy for recurrence?
Round 2
Reviewer 2 Report
Thank you